**Data Availability Statement:** The data underlying the results are available from the UK RENAL

# Sociodemographic features and mortality of individuals on haemodialysis treatment who test positive for SARS-CoV-2: A UK Renal Registry data analysis

Manuela Savino[1]*, Anna Casula[1], Shalini Santhakumaran[1], David Pitcher[1], Esther Wong[1], Winnie Magadi[1], Katharine M. Evans[1], Fran Benoy-Deeney[1], James Griffin[1], Lucy Plumb[1,2], Retha Steenkamp[1], Dorothea Nitsch[1,3,4], James Medcalf[1,5,6]

1 UK Renal Registry, The Renal Association, Bristol, United Kingdom, 2 Population Health Sciences, University of Bristol Medical School, Bristol, United Kingdom, 3 London School of Hygiene and Tropical Medicine, London, United Kingdom, 4 Royal Free London NHS Foundation Trust, London, United Kingdom, 5 University of Leicester, Leicester, United Kingdom, 6 University Hospitals of Leicester NHS Trust, Leicester, United Kingdom

☯ These authors contributed equally to this work.
* Manuela.Savino@renalregistry.nhs.uk

## Abstract

Kidney disease is a recognised risk factor for poor COVID-19 outcomes. Up to 30 June 2020, the UK Renal Registry (UKRR) collected data for 2,385 in-centre haemodialysis (ICHD) patients with COVID-19 in England and Wales. Overall unadjusted survival at 1 week after date of positive COVID-19 test was 87.5% (95% CI 86.1–88.8%); mortality increased with age, treatment vintage and there was borderline evidence of Asian ethnicity (HR 1.16, 95% CI 0.94–1.44) being associated with higher mortality. Compared to the general population, the relative risk of mortality for ICHD patients with COVID-19 was 45.4 and highest in younger adults. This retrospective cohort study based on UKRR data supports efforts to protect this vulnerable patient group.

## Introduction

Coronaviruses (CoVs) are important pathogens in both humans and other vertebrate animals. In their reservoirs, CoV infections mainly affect respiratory, gastrointestinal, liver and central nervous systems [1]. At the end of 2019, a novel coronavirus called Severe Acute Respiratory Syndrome coronavirus 2 (SARS-CoV-2) started spreading across the world, causing a substantive number of cases. The new infectious disease was named COVID-19.

Current evidence shows that comorbidities, such as diabetes, hypertension and chronic kidney disease, and advanced age, are risk factors associated with worse outcomes from COVID-19 [2–4]. Children appear to be less affected than adults [5, 6], although coexisting kidney disease is reported among cases of children requiring intensive care support [7].

A recent meta-analysis reported that patients with COVID-19 developing acute kidney injury had significant 4-fold increased risk of death than patients without acute kidney injury [8].

registry (contact ukrr-research@renalregistry.nhs. uk) for researchers who meet the criteria for access to confidential data.

**Funding:** The author(s) received no specific funding for this work.

**Competing interests:** The authors have declared that no competing interests exist.

In-centre haemodialysis (ICHD) patients regularly visit hospital for their treatment and are at especially high risk, not just because of older age, kidney failure and higher frequency of comorbidities [9], but also because COVID-19 is more likely to spread among hospitalised patients [10]. Moreover, patients with end-stage kidney disease have impairment in both innate and adaptive immunity in uremic state with decreased endocytosis, impaired maturation of monocytes and dendritic cells and malfunction of toll-like receptors [11].

The aim of this retrospective cohort study was to describe the features and mortality of patients on ICHD in England and Wales who had laboratory-confirmed COVID-19.

## Methods

On behalf of the Renal Association, the UK Renal Registry (UKRR) collects patient data without consent under section 251 from the Health Research Authority's Confidentiality Advisory Group. The data were pseudonymised prior to being analysed. This study was approved by the North East Newcastle & North Tyneside 1 Research Ethics Committee (16/NE/0042).

Data on ICHD patients with a positive laboratory test for SARS-CoV-2 were collected by the UK Renal Registry (UKRR) between 26 March and 30 June 2020 using weekly returns from renal centres in England and Wales about each patient (NHS number, date of birth and date of positive test). This collection included all paediatric cases and covered the 18-week period from 3 March to 30 June.

Data were checked and NHS numbers validated, with queries returned to the submitting centre. Using NHS number and date of birth, the Demographics Batch Service was used to retrieve sex, postcode and date of death, if applicable, for every patient in England and Wales. Postcode was used to determine the lower super output area and the associated Index of Multiple Deprivation (IMD) [12].

The Demographics Batch Service is only available in the UK for England and Wales–it was therefore not possible to include in this analysis data from Northern Ireland and Scotland. In addition, Scotland only reports aggregated COVID-19 data to the UK Renal Registry (UKRR).

For survival analysis, patients with a test date after 23 June 2020 were excluded to allow for at least 7 days of follow-up time. Survival time was calculated from the date of positive test to either the date of death or the end of follow-up (30 June). Kaplan Meier unadjusted survival and Cox proportional-hazards models were used to describe survival and hazard ratios. The assumption of proportional hazards was tested and met for the Cox model. To investigate if vintage (time on renal replacement treatment) was associated with an increased risk of mortality, vintage was divided into 3 categories: pre-2016, 2016–2018 and 2019 onwards.

Relative risk of death for English ICHD patients by age and English NHS region were calculated using the England ICHD population (end of 2018, the most recent data available), England mid-year 2018 general population (based on the 2011 census), England ICHD COVID-19 deaths (3 March to 30 June) and England general population COVID-19 deaths (3 March to 30 June).

Analyses were performed using SAS version 9.4. Alpha level of 0.05 and two-sided tests were used throughout.

## Results

At the end of 2018, there were 21,509 adult ICHD patients alive in England and Wales. Between 3 March and 30 June 2020, 50 of 51 renal centres in England and 5 of 5 in Wales reported 2,385 adult ICHD patients with COVID-19 to the UKRR. London had the highest percentage of ICHD patients with COVID-19 (18.7%), with a range of 10.4–24.0% among its 7 renal centres. The remaining regions ranged from 3.5%–12.2% of ICHD patients with

COVID-19. Three COVID-19 cases in ICHD patients aged <18 years were reported from a prevalent paediatric ICHD cohort of 116 (2.6%).

Among adults, most patients with COVID-19 were aged 60–79 years (50.8%) and 18.5% were aged ≥80 years. 62.6% were male, which was just above the proportion of ICHD male patients in the UK (61.8%). A similar proportion of patients that tested positive for SARS-CoV-2 were from ethnic minorities (15.2% were Asian and 11.6% were Black), compared to the overall ICHD population in England and Wales (14.7% Asian and 11.9% Black, respectively) (Table 1).

Survival (unadjusted) at 1 week from the date of a positive SARS-CoV-2 test was 87.5% (95% CI 86.1–88.8%) and 80.0% at 2 weeks (95% CI 78.3–81.5%) (Fig 1).

There were no deaths in children. Patients on ICHD with COVID-19 aged ≥80 years had a mortality risk of about 4.2 times that of those aged 18–59 years (Table 2). There was weak evidence of males with COVID-19 on ICHD doing worse than females (HR 1.19, 95% CI 1.01–1.40). Asian patients on ICHD with COVID-19 had a borderline 16% higher mortality risk than White patients, with no difference seen between Black and White patients. Deprivation was not associated with mortality on ICHD and no interactions were observed. Patients on renal replacement treatment for more than 5 years had a 42% higher mortality risk compared to patients who started dialysis during the last year.

Compared to the general population in England with COVID-19, the relative risk of death in English ICHD patients with COVID-19 was 45.4 (95% CI 41.9–49.1), and decreased with age from a peak in the 20–39 years age group to 9.8 times the risk of death in the general population at ≥80 years (Table 3), with a high interregional variability.

## Discussion

The relative risk of death associated with COVID-19 among ICHD patients was much higher than that of the general population in England, especially among those of younger age. So far,

**Table 1. Demographic comparison between adult patients on ICHD in England and Wales at the end of 2018 and adult patients on ICHD in England and Wales with COVID-19 from 3 March to 23 June 2020.**

| Variable | ICHD adults end 2018 (n = 21,509)[a] | ICHD adults with COVID-19[b] (n = 2,385)[c] |
|---|---|---|
| Age, y | | |
| Median (IQR) | 67 (55–77) | 68 (55–77) |
| 18–39 (%) | 7 | 5 |
| 40–59 (%) | 27 | 26 |
| 60–79 (%) | 48 | 51 |
| ≥80 (%) | 18 | 18 |
| Male (%) | 62 | 63 |
| Ethnicity (%) | | |
| White | 69 | 70 |
| Asian | 15 | 15 |
| Black | 12 | 12 |
| Other | 4 | 3 |
| Most deprived quintile (%) | 30 | 31 |

ICHD, in-centre haemodialysis

[a]Percentages exclude missing data: there were 3% of patients with missing ethnicity and <1% with missing deprivation.

[b]For UK renal centres that submitted patients with COVID-19 to the UKRR.

[c]Percentages exclude missing data: there were 3% with missing ethnicity and 1% with missing deprivation.

[d]Excluded 3 patients aged <18 and 9 patients without at least 1 week follow-up.

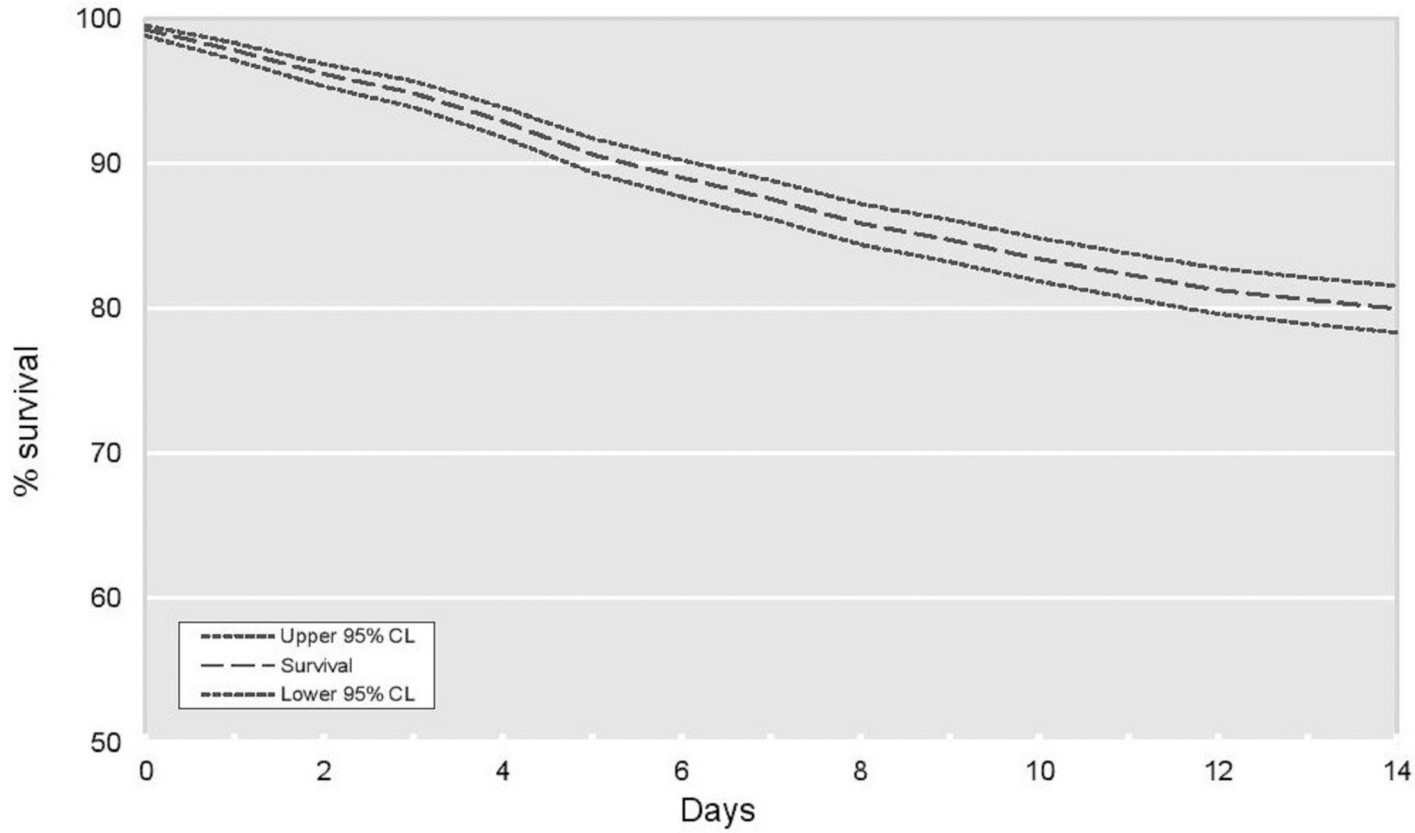

**Fig 1. Kaplan Meier unadjusted survival for Sars-CoV-2 positive patients—2-week survival.**

**Table 2. Multivariable analysis of time to death amongst ICHD adult patients in England and Wales with COVID-19 positive test results, 3 March to 30 June 2020.**

| Variable | No. of patients | No. of deaths | Hazard ratio | 95% CI |
|---|---|---|---|---|
| Age, y | | | | |
| 18–59 | 718 | 86 | 1.00 | |
| 60–79 | 1,190 | 340 | 2.57 | 2.03–3.26 |
| ≥80 | 428 | 187 | 4.24 | 3.28–5.49 |
| Sex | | | | |
| Male | 1,461 | 398 | 1.19 | 1.01–1.40 |
| Female | 875 | 215 | 1.00 | |
| Ethnicity | | | | |
| White | 1,638 | 424 | 1.00 | |
| Asian | 353 | 107 | 1.16 | 0.94–1.44 |
| Black | 277 | 60 | 0.86 | 0.65–1.14 |
| Other | 68 | 22 | 1.36 | 0.89–2.10 |
| Vintage | | | | |
| <2016 | 887 | 276 | 1.42 | 1.15–1.75 |
| 2016–18 | 788 | 196 | 1.12 | 0.90–1.39 |
| ≥2019 | 661 | 141 | 1.00 | |

[a]All variables in the table were mutually adjusted for each other.

[b]Patients with missing ethnicity data were excluded n = 49.

**Table 3. Relative risk of death for ICHD patients with COVID-19 by age group and NHS region in England.**

| Variable | England COVID-19 death rate per 10,000[a] | ICHD COVID-19 death rate per 10,000[b] | Relative risk | |
|---|---|---|---|---|
| | | | ICHD/England COVID-19 death rates | 95% CI |
| Age, y | | | | |
| 20–39 | 0.1 | 60.5 | 432 | 213.4–873 |
| 40–59 | 1.5 | 143.4 | 94.1 | 75.1–118 |
| 60–79 | 10.5 | 346.9 | 33.1 | 29.7–36.9 |
| ≥80 | 55.7 | 548.5 | 9.8 | 8.5–11.3 |
| Region | | | | |
| East of England | 5.4 | 253.2 | 46.7 | 33.5–65.2 |
| London | 6.9 | 433.3 | 63.2 | 55.8–71.7 |
| Midlands | 5.4 | 293.7 | 54.6 | 45.7–65.2 |
| North East & Yorkshire | 5.2 | 267.7 | 51.5 | 40.9–65 |
| North West | 6.5 | 223.7 | 34.5 | 26.1–45.8 |
| South East | 3.7 | 276.5 | 73.9 | 57.3–95.2 |
| South West | 2.3 | 130.6 | 57.7 | 36.8–90.5 |

ICHD, in-centre haemodialysis

[a]Number of deaths recorded 3 March to 30 June 2020 in England divided by the England mid-2018 population estimate.

[b]Number of deaths recorded 3 March to 30 June 2020 in the English ICHD population divided by the end-2018 prevalent English ICHD population.

and in contrast to the adult population, no deaths have been reported in paediatric patients aged <18 years. There was borderline evidence both for males with COVID-19 on ICHD doing worse than females and Asian patients having a 16% higher mortality than White patients.

As observed in other countries, to now, most deaths in the UK have been in people aged ≥50 years, with 55% of these males [13]. Data from the UK Intensive Care National Audit and Research Centre, indicate that among patients with COVID-19 admitted to intensive care, 1.7% had a history of kidney replacement therapy, 70.3% were male and there was an over-representation of those of Asian and Black ethnicities [14], similar to what we report for the ICHD population.

ICHD increases the risk of transmission of infection among patients and the high relative risk of death among younger individuals with COVID-19 on ICHD highlights their extreme vulnerability.

Despite level of deprivation correlating to the percentage of ICHD patients with COVID-19, deprivation was not a major predictor of mortality from COVID-19 on ICHD.

Previous studies have demonstrated the effects of length of time spent on dialysis treatment (vintage) on mortality, concluding that prolonged dialysis increases the mortality risk of patients receiving haemodialysis [15, 16]. A few studies have also shown the specific association between length of time on dialysis and higher risk for infection-related mortality [17, 18].

Moreover, a recent retrospective cohort study in a single centre in Spain analysed the clinical course and outcomes of 36 maintenance haemodialysis patients hospitalised with COVID-19 during the period March to April 2020. The authors concluded that none of the classical cardiovascular risk factors in the general population were associated with higher mortality, and that compared to survivors, non-survivors had significantly longer dialysis vintages [19]. In line with these results, in our study we found a positive association between treatment vintage and survival of ICHD patients with COVID-19.

To better understand the potentially higher mortality from COVID-19 seen for the Asian group, but not in the Black group, other factors such as comorbidities and their effect on survival of COVID-19 patients should be investigated. People of Asian origin are the UK's largest ethnic minority in several cities [20] and type 2 diabetes is up to 6 times more common in Asian than White people [20, 21], which translates to a higher prevalence of diabetes as a cause of requiring ICHD [22].

Although the multivariable analysis was controlled for important factors related to the outcomes, the possibility of residual confounding from unmeasured variables cannot be excluded. We were unable to adjust for comorbidities and/or biochemical data for which both factors detailed comparison data for the general population were not available at the time of our analysis.

Moreover, we could not compare to other treatment modalities, such as peritoneal dialysis or transplantation, because reporting of COVID-19 in these populations is currently unreliable but may be better investigated in future thanks to development of further linkage of UKRR dataset to the national test result data. Data on mortality in ICHD patients as a result of causes not directly related to COVID-19 were not available, but in the future will explain the real burden of mortality associated with COVID-19 among ICHD patients in the UK.

## Conclusions

Despite some limitations, results of this study show the extremely high relative mortality of COVID-19 patients on ICHD and are important to support collective efforts to minimise risk of transmission in this very vulnerable patient group.

## Acknowledgments

We thank staff at all adult and paediatric renal centres in the UK who submit data to the UKRR and who care for kidney patients affected by COVID-19. We are also grateful to the Renal Association COVID-19 National Renal Data Coordinating Group for their guidance on this analysis: Paul Cockwell, Ron Cullen, Rachel Gair, Daniel Gale, Matt Graham-Brown, Thomas Hiemstra, Toby Humphrey, Carol Inward, Rachel Johnson, Graham Lipkin, Fiona Loud, Stephen Marks, Lisa Mumford, Matthew Robb, Adnan Sharif, Neil Sheerin, Laurie Tomlinson and Charlie Tomson.

## Author Contributions

**Conceptualization:** Manuela Savino, Retha Steenkamp, Dorothea Nitsch, James Medcalf.

**Data curation:** Fran Benoy-Deeney, James Griffin, Retha Steenkamp.

**Formal analysis:** Anna Casula, Shalini Santhakumaran, David Pitcher, Esther Wong, Winnie Magadi, Retha Steenkamp.

**Methodology:** Manuela Savino, Anna Casula, Retha Steenkamp, Dorothea Nitsch, James Medcalf.

**Supervision:** Retha Steenkamp, Dorothea Nitsch, James Medcalf.

**Writing – original draft:** Manuela Savino, Anna Casula, Shalini Santhakumaran, David Pitcher, Esther Wong, Winnie Magadi, Katharine M. Evans, Lucy Plumb, Retha Steenkamp.

**Writing – review & editing:** Manuela Savino, Katharine M. Evans, Retha Steenkamp, Dorothea Nitsch, James Medcalf.

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
