## [Decision Letter · Decision Letter 0]

4 Aug 2020

PONE-D-20-21371

Sociodemographic features and mortality of individuals on haemodialysis treatment who test positive for SARS-CoV-2: a UK Renal Registry data analysis

PLOS ONE

Dear Dr. Savino,

Thank you for submitting your manuscript to PLOS ONE. After careful consideration, we feel that it has merit but does not fully meet PLOS ONE’s publication criteria as it currently stands. Therefore, we invite you to submit a revised version of the manuscript that addresses the points raised during the review process.

ACADEMIC EDITOR: I have received the comments of the reviewers on your manuscript. The specific comments of the reviewers are included below. Please provide point by point response in your revised manuscript.

We look forward to receiving your revised manuscript.

Kind regards,

Muhammad Adrish

Academic Editor

PLOS ONE

Journal Requirements:

3. In the ethics statement in the manuscript and in the online submission form, please provide additional information about the patient records used in your retrospective study, including: a) whether all data were fully anonymized before you accessed them; and b) the source of the medical records analyzed in this work (e.g. hospital, institution or registry name).

4. To comply with PLOS ONE submission guidelines, in your Methods section, please provide additional information regarding your statistical analyses. For more information on PLOS ONE's expectations for statistical reporting, please see https://journals.plos.org/plosone/s/submission-guidelines.#loc-statistical-reporting.

Reviewers' comments:

Reviewer's Responses to Questions

**Comments to the Author**

1. Is the manuscript technically sound, and do the data support the conclusions?

Reviewer #1: Partly

Reviewer #2: Yes

Reviewer #3: Yes

2. Has the statistical analysis been performed appropriately and rigorously? 

Reviewer #1: I Don't Know

Reviewer #2: Yes

Reviewer #3: I Don't Know

3. Have the authors made all data underlying the findings in their manuscript fully available?

Reviewer #1: No

Reviewer #2: Yes

Reviewer #3: Yes

4. Is the manuscript presented in an intelligible fashion and written in standard English?

Reviewer #1: Yes

Reviewer #2: Yes

Reviewer #3: Yes

5. Review Comments to the Author

Reviewer #1: This analysis tries to identify factors associated with mortality in a group of 1173 HD patients infected with COVID.

The objective of this study is to evaluate the survival of HD patients after 1 week and 2 weeks of positive diagnosis of SARS-COV-2.

Unadjusted survival at 1 week was 89%. (mortality was 11%)

Unadjusted survival at 2 weeks was 81% (mortality was 19%)

.

The main conclusion is that the relative risk of death associated with COVID -19 in HD patients was much higher that of the general population in England.

What were the factors associated with COVID infection?

The authors analyzed only age, gender and ethnicity.

For a global analysis of mortality, in a multivariable analysis, other factors that are classically associated with higher risk of death in dialysis patients, should also have been included: time on dialysis, diabetes, albumin, CRP, phosphate levels, etc.

In my opinion, though interesting, this analysis is merely descriptive.

I had not access to the Kaplan Meyer curve.

Reviewer #2: Technically sound well written with appropriate statistical analysis. It would have been interesting to know the association of diabetes as it effected various socio-demographic groups and whether is was the main a factor for high mortality in Asian population.

Reviewer #3: Author has made an reasonable attempt to publish mortality of COVID-19 in Incentre hemodialysis patients using UKRR which has not been published before. Overall the paper reads well but I would recommend a few suggestions as below.

1. Page 3 under Introduction, 2nd paragraph please add "A recent meta-analysis reported that patients with COVID-19 developing acute kidney injury had significant 4-fold increased risk of death than patients without acute kidney injury".

Please cite paper: Survival rate in acute kidney injury superimposed COVID-19 patients: a systematic review and meta-analysis. Renal Failure. 2020 Jan 1;42(1):393-7.

2. It would be good to add in the introduction that patients with ESRD have relative immunodeficiency. I would recommend adding the following sentence to Page 3 under introduction, Paragraph 3. "ESRD patients have impairment in both innate and adaptive immunity in uremic state with decreased endocytosis, impaired maturation of monocytes and dendritic cells and malfunction of toll-like receptors contributing to relative immunodeficiency"

Please cite paper: Potential role of plasmapheresis in severe cytomegalovirus infection with ongoing immunemediated hemolysis and low complement level. Journal of Renal Injury Prevention. 2017 Sep 8;7(3):206-10.

3. Why do we think asian patients have higher mortality. I see that you have explained in the paper to a certain extent. But curious to know what age group would asians mostly fit into? Was obesity a known factor in these patients? I see that asians mostly had diabetes which by itself is a higher comorbid condition. I hope we are not biased by the fact that there is missing data on ethnicity of 22% of patients and overestimating the effect or the fact that we were unable to adjust for comorbidities?

4. Completely agree that higher age >80 mortality is very high and is a good point highlighted in the paper.

5. Would recommend adding a few lines of Conclusion after discussion part highlighting points.

6. PLOS authors have the option to publish the peer review history of their article (what does this mean?). If published, this will include your full peer review and any attached files.

Reviewer #1: **Yes: **Teresa Adragão

Reviewer #2: **Yes: **Aasim Ahmad

Reviewer #3: **Yes: **Sohail Abdul Salim

---

## [Author Response · Author response to Decision Letter 0]

21 Sep 2020

Comments Reviewer 1:

1) This analysis tries to identify factors associated with mortality in a group of 1173 HD patients infected with COVID.

The objective of this study is to evaluate the survival of HD patients after 1 week and 2 weeks of positive diagnosis of SARS-COV-2.

Unadjusted survival at 1 week was 89%. (mortality was 11%)

Unadjusted survival at 2 weeks was 81% (mortality was 19%)

The main conclusion is that the relative risk of death associated with COVID -19 in HD patients was much higher that of the general population in England.

What were the factors associated with COVID infection?

The authors analyzed only age, gender and ethnicity.

Author's response:

We thank the reviewer for the thorough review of our paper and believe her input has been important in making our paper more balanced. 

The reviewer’s comment raises the important point related to factors associated with Covid-19 infection. While we agree that other risk factors are important, we do not have comparative data for the general population, because these were not reported to Public Health England at the time of our analysis. Therefore, even if we had had data on other factors to include, such as comorbidities, we could not have used them for a comparison with the general population. Additionally, our data on comorbidities for the ICHD population are currently incomplete and need linkage to hospital records to augment them. 

We have highlighted this issue in the discussion.

2) For a global analysis of mortality, in a multivariable analysis, other factors that are classically associated with higher risk of death in dialysis patients, should also have been included: time on dialysis, diabetes, albumin, CRP, phosphate levels, etc.

In my opinion, though interesting, this analysis is merely descriptive.

Author's response: We agree that our analysis is descriptive, but nevertheless it was instrumental in prompting the UK government to change policy on shielding of dialysis patients, because of the stark difference in mortality between the ICHD and general populations. 

This revised version of our paper now also includes a much larger cohort of 2,385 ICHD patients up to 30 June 2020 and our findings confirm a striking higher relative risk of mortality (45.4 times higher) of dialysis patients compared to the general population (this change is tracked in both the methods and Table 3 in results). We therefore think that our analysis has an important contribution to the epidemiological question regarding whether dialysis patients are different in risk of mortality compared to the general population. 

Among the additional factors listed associated with higher risk of death in dialysis patients, we could only adjust our analysis for dialysis vintage and found a positive association. This is now highlighted in the methods sessions and in the results session (Table 2). The importance of these findings has been highlighted in the discussion too. We have highlighted in the discussion possible limitations related to remaining unmeasured confounders.

3) I had not access to the Kaplan Meyer curve.

Author's response: Thank you for the important point raised about the Kaplan-Meier curve which is now available as Figure 1 in the results.

Comments Reviewer 2:

1) Technically sound well written with appropriate statistical analysis.

Author' s response: We thank the reviewer for the thoughtful review of our work and kind words.

2) It would have been interesting to know the association of diabetes as it effected various socio-demographic groups and whether is was the main a factor for high mortality in Asian population.

Author's response: Unfortunately, we did not have a break down for diabetes status and ethnicity for the general population as at the time of our analysis these data were not available from Public Health England. Moreover, our data on dialysis patients' diabetes status are currently incomplete and require linkage with hospital records, which is being addressed but beyond the scope of this paper. We have highlighted this issue in our discussion.

Comments Reviewer 3:

1) Author has made a reasonable attempt to publish mortality of COVID-19 in Incentre hemodialysis patients using UKRR which has not been published before. Overall, the paper reads well but I would recommend a few suggestions as below.

1. Page 3 under Introduction, 2nd paragraph please add "A recent meta-analysis reported that patients with COVID-19 developing acute kidney injury had significant 4-fold increased risk of death than patients without acute kidney injury".

Please cite paper: Survival rate in acute kidney injury superimposed COVID-19 patients: a systematic review and meta-analysis. Renal Failure. 2020 Jan 1;42(1):393-7.

2. It would be good to add in the introduction that patients with ESRD have relative immunodeficiency. I would recommend adding the following sentence to Page 3 under introduction, Paragraph 3. "ESRD patients have impairment in both innate and adaptive immunity in uremic state with decreased endocytosis, impaired maturation of monocytes and dendritic cells and malfunction of toll-like receptors contributing to relative immunodeficiency"

Please cite paper: Potential role of plasmapheresis in severe cytomegalovirus infection with ongoing immunemediated hemolysis and low complement level. Journal of Renal Injury Prevention. 2017 Sep 8;7(3):206-10.

Author's response: We thank the reviewer for his thoughtful and thorough review. We have thoroughly re-reviewed the manuscript and accepted his suggestions. Both citations were added to the introduction.

2) Why do we think Asian patients have higher mortality? I see that you have explained in the paper to a certain extent. But curious to know what age group would Asians mostly fit into? 

Author's response: We thank the reviewer for the points and observations raised. The variables included in the multivariable model are adjusted for each other. Additionally, to respond to the point raised, while we know that the total Asian prevalent ICHD population is younger compared to the White population, we found that in the COVID-19 positive cohort the median age of the Asian group is the same as that of the White group. We attach to this response letter supplementary tables related to these analyses.

3) Was obesity a known factor in these patients? 

Author's response: A primary aim was to compare the risk between dialysis patients and the general population and general population data we had access to did not report ethnicity or BMI. Moreover, BMI data at dialysis start are incomplete. BMI is known to not show the usual associations with death on dialysis [1] so we believe that it is unlikely to explain associations seen.

1. Kalantar-Zadeh K, Rhee CM, Chou J, et al. The Obesity Paradox in Kidney Disease: How to Reconcile it with Obesity Management. Kidney Int Rep. 2017;2(2):271-281. doi:10.1016/j.ekir.2017.01.009

4) I see that Asians mostly had diabetes which by itself is a higher comorbid condition. I hope we are not biased by the fact that there is missing data on ethnicity of 22% of patients and overestimating the effect or the fact that we were unable to adjust for comorbidities?

Author's response: For comparisons of those on ICHD with and without Covid-19 these more detailed analyses are beyond the scope of this paper as we require linkage to hospital records to address missing comorbidity data. 

This analysis is descriptive, and we did not explore comorbidities in detail. In this revised version of our paper we have also updated the cohort to the 30th of June including 2,385 ICHD patients and reduced missing ethnicity data in the previous cohort (from 22% to 3%). Findings from the updated version confirm a striking higher relative risk of mortality of dialysis patients compared to the general population of 45.4 and still confirm a borderline association between the Asian group and risk of mortality which is 16% higher than the risk of the White group.

5) Completely agree that higher age >80 mortality is very high and is a good point highlighted in the paper.

Author's response: We thank the reviewer for his support.

6) Would recommend adding a few lines of Conclusion after discussion part highlighting points.

Author's response: We thank the reviewer for this suggestion and added a Conclusion section to highlight the key points of our manuscript.

---

## [Decision Letter · Decision Letter 1]

13 Oct 2020

Sociodemographic features and mortality of individuals on haemodialysis treatment who test positive for SARS-CoV-2: a UK Renal Registry data analysis

PONE-D-20-21371R1

Dear Dr. Savino,

We’re pleased to inform you that your manuscript has been judged scientifically suitable for publication and will be formally accepted for publication once it meets all outstanding technical requirements.

Kind regards,

Muhammad Adrish

Academic Editor

PLOS ONE

Additional Editor Comments (optional):

You have satisfactorily answered all the queries that have been raised by the reviewers.

Reviewers' comments:

Reviewer's Responses to Questions

**Comments to the Author**

1. If the authors have adequately addressed your comments raised in a previous round of review and you feel that this manuscript is now acceptable for publication, you may indicate that here to bypass the “Comments to the Author” section, enter your conflict of interest statement in the “Confidential to Editor” section, and submit your "Accept" recommendation.

Reviewer #2: All comments have been addressed

Reviewer #3: All comments have been addressed

2. Is the manuscript technically sound, and do the data support the conclusions?

Reviewer #2: Yes

Reviewer #3: Yes

3. Has the statistical analysis been performed appropriately and rigorously? 

Reviewer #2: Yes

Reviewer #3: Yes

4. Have the authors made all data underlying the findings in their manuscript fully available?

Reviewer #2: Yes

Reviewer #3: Yes

5. Is the manuscript presented in an intelligible fashion and written in standard English?

Reviewer #2: Yes

Reviewer #3: Yes

6. Review Comments to the Author

Reviewer #2: Thank you for addressing our comments and suggestions and explaining the reasons for those that could not be addressed

Reviewer #3: Thank you for addressing the concerns and incorporating my suggestions into the paper. Paper reads well at this time.

7. PLOS authors have the option to publish the peer review history of their article (what does this mean?). If published, this will include your full peer review and any attached files.

Reviewer #2: **Yes: **Professor Aasim Ahmad

Reviewer #3: No

---

## [Editor Report · Acceptance letter]

16 Oct 2020

PONE-D-20-21371R1 

Sociodemographic features and mortality of individuals on haemodialysis treatment who test positive for SARS-CoV-2: a UK Renal Registry data analysis 

Dear Dr. Savino:

I'm pleased to inform you that your manuscript has been deemed suitable for publication in PLOS ONE. Congratulations! Your manuscript is now with our production department. 

Kind regards, 

on behalf of

Dr. Muhammad Adrish 

Academic Editor

PLOS ONE